# On Relating Explanations and Adversarial Examples

**Alexey Ignatiev**
Monash University, Australia
`alexey.ignatiev@monash.edu`

**Nina Narodytska**
VMWare Research, CA, USA
`nnarodytska@vmware.com`

**Joao Marques-Silva**
ANITI, Toulouse, France
`joao.marques-silva@univ-toulouse.fr`

## Abstract

The importance of explanations (XP's) of machine learning (ML) model predictions and of adversarial examples (AE's) cannot be overstated, with both arguably being essential for the practical success of ML in different settings. There has been recent work on understanding and assessing the relationship between XP's and AE's. However, such work has been mostly experimental and a sound theoretical relationship has been elusive. This paper demonstrates that explanations and adversarial examples are related by a generalized form of hitting set duality, which extends earlier work on hitting set duality observed in model-based diagnosis and knowledge compilation. Furthermore, the paper proposes algorithms, which enable computing adversarial examples from explanations and vice-versa.

## 1 Introduction

Adversarial examples (AE's) [54] illustrate the brittleness of machine learning (ML) models, and have been the subject of growing interest in recent years. Explanations (XP's) of (black-box) ML models provide trust in ML models, and exemplify the increasing importance of eXplainable AI (XAI) [17, 12, 13]. Over the last few years, a number of works realized the existence of some connection between AE's and XP's [34, 55, 48, 56, 59, 41, 8]. However, past work has been experimental, and a deeper theoretical connection between AE's and XP's has been elusive. This paper demonstrates the existence of such a theoretical connection between AE's and XP's.

In this work we take a formal logic point of view on the analysis of ML models. Namely, we employ first order logic (FOL) as a framework to specify an ML model, define notions of an explanation and a counterexample to that explanation, which can be viewed as a generalization of an adversarial example. We then demonstrate that these notions possess well-known counterparts in the FOL terminology, like prime implicants and implicates, respectively. Such formalization allows us to obtain our main result that reveals a duality relation between explanations and counterexamples. Based on this connection, we show how explanations and counterexample interact. For example, explanations can be used to generate counterexamples. Dually, we can generate explanations given counterexamples. Furthermore, we also show how to compute adversarial examples from counterexamples. The ideas in the paper build on tightly related work in model-based diagnosis [45], namely the hitting set duality between diagnoses and conflicts, but also builds on related work in knowledge compilation, concretely in the use of prime implicants and implicates to compile knowledge [51, 36].

The paper is organized as follows. Section 2 introduces concepts used in the remainder of the paper. Section 3 investigates the connections between adversarial examples and explanations, and proposes algorithms for the enumeration of explanations and adversarial examples. Experimental

evidence demonstrating the relationship between explanations and adversarial examples is analyzed in Section 4. The paper concludes in Section 5.

## 2  Background

### 2.1  Preliminaries

We consider an ML model $\mathbb{M}$, represented by a finite set of first order logic (FOL) sentences $\mathcal{M}$. (Where viable, alternative representations for $\mathcal{M}$ can be considered, e.g. fragments of FOL, (mixed-)integer linear programming, constraint language(s), etc.) A set of features $\mathcal{F} = \{f_1, \ldots, f_L\}$ is assumed. Each feature $f_i$ is categorical (or ordinal), with values taken from some set $D_i$. An *instance* is an assignment of values to features. The space of instances, also referred to as *feature* (or *instance*) *space*, is defined by $\mathbb{F} = D_1 \times D_2 \times \ldots \times D_L$. (Domains may or may not have an order relation. Also, for real-valued features, a suitable interval discretization can be considered.) A (feature) literal $\lambda_i$ is of the form $(f_i = v_i)$, with $v_i \in D_i$. In what follows, a literal will be viewed as an atom, i.e. it can take value *true* or *false*. As a result, an instance can be viewed as a set of $L$ literals, denoting the $L$ distinct features, i.e. an instance contains a single occurrence of a literal defined on any given feature. A set of literals is consistent if there exists at most one literal associated with each feature. A consistent set of literals can be interpreted as a conjunction or as a disjunction of literals; this will be clear from the context. When interpreted as a conjunction, the set of literals denotes a *cube* in instance space, where the unspecified features can take any possible value of their domain. When interpreted as a disjunction, the set of literals denotes a *clause* in instance space. As before, the unspecified features can take any possible value of their domain.

The remainder of the paper assumes a classification problem with a set of classes $\mathbb{K} = \{\kappa_1, \ldots, \kappa_M\}$. A prediction $\pi \in \mathbb{K}$ is associated with each instance $\mathcal{I}$.

Given some target prediction $\pi \in \mathbb{K}$, one can devise representations for the formula $F_{\mathcal{M},\pi} \triangleq (\mathcal{M} \to \pi)$ [52, 23]. In particular, we will be interested in computing prime implicants and implicates of $F_{\mathcal{M},\pi}$, where a consistent set of feature literals $\tau$ is an implicant of $F_{\mathcal{M},\pi}$ if $\tau \vDash F_{\mathcal{M},\pi}$, and a consistent set of feature literals $\nu$ is a (negated) implicate of $F_{\mathcal{M},\pi}$ if $F_{\mathcal{M},\pi} \vDash \neg\nu$, or alternatively $(\nu \vDash \neg F_{\mathcal{M},\pi}) \equiv (\nu \vDash \vee_{\rho \neq \pi}(\mathcal{M} \to \rho))$. An implicant $\tau$ (implicate $\nu$, resp.) is called *prime* if none of its proper subsets $\tau' \subsetneq \tau$ ($\nu' \subsetneq \nu$, resp.) is an implicant (implicate, resp.).

Throughout the paper, it will be convenient to use a more detailed notation, where ML models, prime implicants and prime implicates represent functions, respectively mapping $\mathbb{F}$ into $\mathbb{K}$ and $\{0, 1\}$. Concretely, the ML model $\mathbb{M}$ computes a function $\mathcal{M} : \mathbb{F} \to \mathbb{K}$ [1]. As a result, given some instance $X \in \mathbb{F}$ in feature space, $\mathcal{M}(X)$ denotes the prediction computed by the ML model. Furthermore, the notation $\tau \vDash F_{\mathcal{M},\pi}$ represents the following first order logic statement:

$$\forall(X \in \mathbb{F}).\tau(X) \to (\mathcal{M}(X) = \pi) \tag{1}$$

where $\tau$ is a Boolean function mapping $\mathbb{F}$ into $\{0, 1\}$, and $\mathcal{M}$ is a function mapping $\mathbb{F}$ into $\mathbb{K}$. Essentially, a prime implicant is viewed as a Boolean function taking value 1 for a *cube* (i.e. set of points) in feature space for which the prediction is $\pi$. Similarly, the notation $\nu \vDash \neg F_{\mathcal{M},\pi}$ represents the following first order logic statement:

$$\forall(X \in \mathbb{F}).\nu(X) \to (\vee_{\rho \neq \pi}\mathcal{M}(X) = \rho) \tag{2}$$

where $\nu$ is a Boolean function mapping $\mathbb{F}$ into $\{0, 1\}$.

**Example 1.** *The paper's running example is the restaurant example from Russell&Norvig's book [50, Fig. 18.3, page 700]. For this example, the set of features is:*

$\{\mathsf{A(lternate)}, \mathsf{B(ar)}, \mathsf{W(eekend)}, \mathsf{H(ungry)}, \mathsf{Pa(trons)}, \mathsf{Pr(ice)}, \mathsf{Ra(in)}, \mathsf{Re(serv.)}, \mathsf{T(ype)}, \mathsf{E(stim.)}\}.$

*For instance,* A, B, W, H, Ra, Re *are Boolean features taking* True *or* False *values.* T *is a categorical feature with four possible values* {Burger, French, Italian, Thai}. *The other domains are defined similarly. The dataset predicts whether the customer should wait or not in a given situation. So we have the target label* Wait *that takes* Yes *or* No *values. An example instance is:* $\{\mathsf{A}, \neg\mathsf{B}, \neg\mathsf{W}, \mathsf{H}, (\mathsf{Pa} = \mathsf{Some}), (\mathsf{Pr} = \$\$\$), \neg\mathsf{Ra}, \mathsf{Re}, (\mathsf{T} = \mathsf{French}), (\mathsf{E} = \mathsf{0\text{–}10})\}.$

**Example 2.** *Throughout the paper, the examples will consider decision sets [28]. (The selection of decision sets is motivated by simplicity. The actual experiments consider black-box models.) For the example dataset, a decision set obtained with an off-the-shelf tool is:*

| | | | | | |
|---|---|---|---|---|---|
| **IF** | (Pa = Some) $\wedge \neg$(E = >60) | **THEN** | (Wait = Yes) | | (R1) |
| **IF** | W $\wedge \neg$(Pr = \$\$\$) $\wedge \neg$(E = >60) | **THEN** | (Wait = Yes) | | (R2) |
| **IF** | $\neg$W $\wedge \neg$(Pa = Some) | **THEN** | (Wait = No) | | (R3) |
| **IF** | (E = >60) | **THEN** | (Wait = No) | | (R4) |
| **IF** | $\neg$(Pa = Some) $\wedge$ (Pr = \$\$\$) | **THEN** | (Wait = No) | | (R5) |

## 2.2 Related Work

We overview two main research directions: (a) methods for generating adversarial attacks and (b) methods for producing explanations of ML model decisions. These two research directions have similarities, e.g. both types of methods make assumptions about transparency of the model, i.e. whether it is a white-box or a black-box, enforce different guarantees on the outcome (best effort vs guaranteed solution), etc. However, somewhat surprisingly, a formal connection between adversarial examples and explanations has not been proposed in the literature. Our work bridges this gap.

**Adversarial attacks.** Szegedy *et al.* demonstrated that ML models lack robustness: a small perturbation of an input may lead to a significant perturbation of the output of an ML model [54]. This vulnerability can be exploited to augment the original input with a crafted perturbation, invisible to a human but sufficient for the ML model to misclassify this input. A perturbation is required to be small w.r.t. a given metric, e.g. $l_1, l_2$, and $l_\infty$ [54, 16, 6] norms. This influential work triggered several new research directions [7].

Depending on the threat model, adversarial attack generators can be broadly partitioned into black-box and white-box methods. Black-box methods assume that the attacker has no knowledge about the ML model. In this case, adversarial attacks are based on algorithms that recover the original model structure and transfer adversarial examples to the original model [42, 43, 27]. In contrast, white-box methods assume that the adversary has complete knowledge about the model, e.g. [54, 6, 37, 38]. In this work, we lean towards a black-box model, however we assume the existence of a logic-based oracle that can be queried about entailment relations between inputs and outputs. The majority of white-box methods to produce adversarial examples are heuristic and rely on gradient descent methods. Hence, they cannot guarantee that an adversarial example will be found even if it exists. For safety-critical applications such uncertainty might not be tolerable, therefore, a new trend is emerging focusing on methods with *provable guarantees* [25, 4, 33, 40, 10, 14, 26, 30]. For example, Katz *et al.* proposed the Reluplex system that finds an adversarial example if it exists in ReLU-based networks [25]. The main idea is to encode a network function as an SMT formula and to prove its properties (e.g. the absence of an adversarial perturbation in a given neighborhood) using an SMT and ILP hybrid. A formal approach was also applied to binarized neural networks [11, 40, 10, 26]. Finally, there is work on understanding adversarial examples. Xu *et al.* provide sensitivity analysis of pixel level perturbations and investigate the effect of these perturbations on internal layers of the network [58]. In [59], the authors produce structured attacks, where the attack mechanism achieves strong group sparsity leading to more interpretable examples.

**Model explanations.** Explainability of ML models depends on the type of the model that a user works with. There is a class of models considered to be interpretable by a human decision maker, like decision trees, lists or sets. When considering interpretable ML models, the goal is to compute models that provide *minimal* explanations associated with each prediction [28, 2, 39, 24, 49][2].

In case of non-interpretable models, like neural networks or ensembles of trees, there are two main options: (a) recompile (augment) the original model into (with) an explainable model [15, 57] or (b) extract an explanation from the model. The former approach might not be suitable if we want to provide explanations with guarantees w.r.t. the original model. The latter approach is currently the mainstream one. As in the case of adversarial attacks, the method designer needs to make an

assumption on transparency of the model to the explainer. As above, white-box methods rely on computing gradients, e.g. saliency maps or integrated gradients [53]. However, these methods are mostly applicable to computer vision tasks. One influential line of research looks into explaining black-box models [19]. Explanations can be local [46, 35, 18] or global [47, 29], depending on whether they only apply to a local neighborhood of a target instance or not. While these methods can provide probabilistic guarantees, they do not provide worst-case guarantees on generated explanations. Moreover, there exist concerns regarding robustness of some of these methods [1].

Recently, two approaches were proposed to compute global explanations. The first method takes a compilation-based approach to computing global explanations [52]. If it is possible to compile an ML model to a suitable compilation structure, this method can extract all possible global explanations. However, the main drawback of this approach is exponential worst-case size of the compiled representation. In [23], the authors proposed new methods for computing explanations, by extracting prime implicants. This approach scales better compared to the compilation based approach and can generate a number of global explanations on demand. The current work is based on ideas from [23] to generate explanations and counterexamples.

There is a recent line of work on defending ML models against adversarial attacks based on interpretablity [34, 55]. For example, in [55] the authors identify neurons that correspond to human perceptible attributes and check whether these attributes are used in classification of the input. If so the input is non-adversarial and adversarial otherwise. Tomsett *et al.* [56] stated that adversarial examples and explanations are related notions. Namely, they argue that adversarial examples can improve ML interpretability and vice versa, e.g. neurons activations patterns are different for adversarial and original inputs which provides an insight about the network's internal representation. Finally, there is work on using advanced training procedures, like robust training, to improve the network interpretablity [48, 41, 8]. For example, in [48] the authors propose to regularize input gradients to improve robustness to transferred adversarial examples and quality of gradient-based explanations. Chalasani *et al.* proposed to employ adversarial training to learn logistic models with the feature-concentration property that are easier for the user to interpret [8].

# 3 Relating Explanations and Adversarial Examples

The goal of this section is to establish a tight connection between adversarial examples and explanations. To achieve this goal, we must first formalize the notion of (absolute) explanations and that of counterexample, and prove a (minimal) hitting set relationship between the two. Afterwards, we demonstrate how adversarial examples can be computed from explanations and vice-versa.

## 3.1 Explanations & Counterexamples

In this section, we introduce two new notions: *absolute explanations* and *counterexamples* over the input features. An absolute explanation for the prediction $\pi$ is a generalization of commonly used local and global explanations [46, 47, 52, 23]. An absolute explanation is the strongest form of an explanation that does not depend on a concrete input instance and acts globally over the entire feature space. For any instance that *matches* an absolute explanation, the ML model prediction is guaranteed to be $\pi$. By matching, we mean that a set of features shared by the instance and the explanation have the same values. The second notion is a counterexample to the prediction $\pi$, which is a generalization of commonly used adversarial and some forms of universal adversarial examples [46, 37]. Intuitively, a counterexample is a set of input feature values that forces the ML model to output a prediction that is different from $\pi$. We are mostly interested in minimal such sets. Again, it is a strong notion as any instance that matches the counterexample must not be classified as $\pi$.

Given an ML model, represented by some logic encoding $\mathcal{M}$, and a prediction $\pi \in \mathbb{K}$, the following definitions are considered.

**Definition 1** (Explanation). *A(n absolute) explanation (XP) of a prediction $\pi$ is a subset-minimal set[3] of literals $\mathcal{E}$, representing distinct features, such that $\mathcal{E} \vDash (\mathcal{M} \to \pi)$.*

Observe that explanations are often deemed as local [46, 47]. Alternatively, *global* explanations (although hold in the complete instance space) are relative to an instance $\mathcal{I}$, when $\mathcal{E} \subseteq \mathcal{I}$ [23]. Explanations in this paper are independent of a concrete instance, and so are referred to as *absolute*.

**Definition 2** (Counterexample)**.** *A subset-minimal set $\mathcal{C}$ of literals is a* counterexample *(CEx) to a prediction $\pi$, if $\mathcal{C} \vDash (\mathcal{M} \to \rho)$, with $\rho \in \mathbb{K} \wedge \rho \neq \pi$.*

Clearly, an explanation $\mathcal{E}$ is a prime implicant of $F_{\mathcal{M},\pi}$ and a counterexample $\mathcal{C}$ is a (negated) prime implicate of $F_{\mathcal{M},\pi}$.

**Example 3.** *For the running example, due to* (R1)*, an explanation for the prediction* (Wait $=$ Yes) *is:* (Pa $=$ Some) $\wedge \neg$(E $= >60$)*. Moreover, due to* (R5)*, a counterexample for the prediction* (Wait $=$ Yes) *is:* $\neg$(Pa $=$ Some) $\wedge$ (Pr $=$ \$\$\$)*.*

Two literals are *inconsistent* if they represent the same feature but refer to different values. We say that a literal $\tau_i$ *breaks* a set of literals $\mathcal{S}$ (each denoting a different feature) if $\mathcal{S}$ contains a literal inconsistent with $\tau_i$. Thus we can talk about breaking an explanation $\mathcal{E}$ or breaking a counterexample $\mathcal{C}$. Moreover, two sets of literals $\mathcal{S}_1$ and $\mathcal{S}_2$ break each other if they contain literals in the same feature referring to different values.

**Example 4.** *For the running example, the explanation corresponding to the set of literals $\mathcal{S}_1 = \{$(Pa $=$ Some)$, \neg$(E $= >60$)$\}$ breaks the counterexample corresponding to the set of literals $\mathcal{S}_2 = \{\neg$(Pa $=$ Some)$,$ (Pr $=$ \$\$\$)$\}$ and vice-versa, as* (Pa $=$ Some) *and* $\neg$(Pa $=$ Some) *are the inconsistent literals in this case.*

As hinted in the examples above, we can now state the paper's main result. We start with a general assumption.

**Assumption 1.** *The ML model $\mathbb{M}$ computes a function $\mathcal{M} : \mathbb{F} \to \mathbb{K}$.*

This assumption is essential to ensure that for any instance in feature space the prediction is unique.

**Theorem 1.** *Given an ML model $\mathbb{M}$, represented by some logic encoding $\mathcal{M}$, and a prediction $\pi$, every explanation $\mathcal{E}$ of $\pi$ breaks every counterexample of $\pi$, and every counterexample $\mathcal{C}$ of $\pi$ breaks every explanation of $\pi$.*

*Proof.* The proof of the theorem statement consists of two parts:

1. Every explanation $\mathcal{E}$ of $\pi$ breaks every counterexample $\mathcal{C}$ of $\pi$.

   | | |
   |---|---|
   | $\forall(X \in \mathbb{F}).\mathcal{C}(X) \to (\vee_{\rho \neq \pi} \mathcal{M}(X) = \rho)$ | Definition of some counterexample $\mathcal{C}$ |
   | $\forall(X \in \mathbb{F}).\neg(\vee_{\rho \neq \pi} \mathcal{M}(X) = \rho) \to \neg\mathcal{C}(X)$ | Contrapositive |
   | $\forall(X \in \mathbb{F}).(\mathcal{M}(X) = \pi) \to \neg\mathcal{C}(X)$ | Negation given set of classes |
   | $\forall(X \in \mathbb{F}).\mathcal{E}(X) \vDash (\mathcal{M}(X) = \pi)$ | Definition of some explanation $\mathcal{E}$ |
   | $\forall(X \in \mathbb{F}).\mathcal{E}(X) \vDash \neg\mathcal{C}(X)$ | Explanation breaks counterexample |

2. Every counterexample $\mathcal{C}$ of $\pi$ breaks every explanation $\mathcal{E}$ of $\pi$.

   | | |
   |---|---|
   | $\forall(X \in \mathbb{F}).\mathcal{E}(X) \to (\mathcal{M}(X) = \pi)$ | Definition of some explanation $\mathcal{E}$ |
   | $\forall(X \in \mathbb{F}).\neg(\mathcal{M}(X) = \pi) \to \neg\mathcal{E}(X)$ | Contrapositive |
   | $\forall(X \in \mathbb{F}). \vee_{\rho \neq \pi} (\mathcal{M}(X) = \rho) \to \neg\mathcal{E}(X)$ | Negation given set of classes |
   | $\forall(X \in \mathbb{F}).\mathcal{C}(X) \vDash \vee_{\rho \neq \pi}(\mathcal{M}(X) = \rho)$ | Definition of some counterexample $\mathcal{C}$ |
   | $\forall(X \in \mathbb{F}).\mathcal{C}(X) \vDash \neg\mathcal{E}(X)$ | Counterexample breaks explanation |

   $\square$

As argued below, by listing all minimal counterexamples, we can extract all minimal explanations, and vice-versa. Furthermore, one can readily conclude that Theorem 1 generalizes the hitting set relationship between diagnoses and conflicts first investigated by Reiter in the 80s [45], and since then studied in different settings [5, 3, 32].

**Example 5** (Duality)**.** *For the running example, let the prediction again be* (Wait $=$ Yes) *and the decision set proposed in Example 2. For this prediction and given the ML model, there are two global explanations:*

   1. (Pa $=$ Some) $\wedge \neg$(E $= >60$)*; and*
   2. W $\wedge \neg$(Pr $=$ \$\$\$) $\wedge \neg$(E $= >60$)*.*

*This means that as long as any of these two conjunctions of literals holds, then the prediction will be* (Wait $=$ Yes)*. Moreover, there are three counterexamples (i.e. explanations for not predicting* (Wait $=$ Yes) *(which for this example corresponds to predicting* (Wait $=$ No)*):*

*1.* $\neg W \land \neg(\mathsf{Pa} = \mathsf{Some})$*;*
*2.* $(\mathsf{E} = {>}60)$*; and*
*3.* $\neg(\mathsf{Pa} = \mathsf{Some}) \land (\mathsf{Pr} = \$\$\$)$*.*

*This means that as long as any of these three conjunctions of literals holds, then the prediction will* **not** *be* $(\mathsf{Wait} = \mathsf{Yes})$*. It can be readily concluded that the XP's (minimally) break the CEx's and vice-versa.*

**Remark 1.** *As hinted in Example 5, and building on earlier work on model-based diagnosis and enumeration of prime implicants and implicates [45, 51, 44], it is straightforward to compute counterexamples from explanations and vice-versa:*

*1. If we have the set of explanations for a prediction, then we can compute the set of counterexamples as the* consistent *minimal breaks of the set of explanations.*
*2. Similarly, if we have the set of counterexamples, then we can compute the set of explanations as the* consistent *minimal breaks of the set of counterexamples.*

**Remark 2.** *The assumptions for relating explanations with counterexamples are fairly general. For example, features do not need to be ordinal. Clearly, adversarial examples expect features to be ordinal. This is covered in the next section.*

## 3.2 Relationship with Adversarial Examples

The previous section showed that each explanation breaks every counterexample, and that each counterexample breaks every explanation. This holds for any machine learning model for which the logic representation of the model computes a function mapping feature space $\mathbb{F}$ into $\mathbb{K}$. Adversarial examples were introduced in earlier work [54] *denoting small changes to the features w.r.t. to a given distance measure that results in the prediction error*. In this paper, we use the following definition of adversarial example. Given an instance $\mathcal{I}$ in feature space, corresponding to prediction $\pi$, our goal is to find another instance, $\mathcal{I}_{\mathsf{ae}}$, corresponding to a different prediction, and which is *closest* to the original instance. Formally,

$$
\begin{aligned}
\min \quad & \mathrm{Dist}(\mathcal{I}_{\mathsf{ae}}, \mathcal{I}) \\
\mathrm{st} \quad & \mathcal{I}_{\mathsf{ae}} \models \vee_{\rho \neq \pi} \rho
\end{aligned} \tag{3}
$$

Clearly, adversarial examples assume that features are ordinal, enabling the notion of distance to be well-defined. (For real-valued features, we assume an often-used discretization of the input.) We can now relate adversarial examples with counterexamples and with explanations.

**Theorem 2** (From XP's to AE's)**.** *Given an ML model* $\mathbb{M}$*, represented with a logic representation* $\mathcal{M}$*, a prediction* $\pi$*, with set of explanations* $\mathbb{E}$ *and set of counterexamples* $\mathbb{C}$*, and an instance* $\mathcal{I}$ *taken from feature space, let* $\mathcal{C}_{\mathsf{ae}}$ *denote the counterexample with minimum distance to* $\mathcal{I}$*. Then* $\mathcal{I}_{\mathsf{ae}}$ *corresponds to* $\mathcal{C}_{\mathsf{ae}}$ *by setting the unspecified feature values to the values in* $\mathcal{I}$*.*

*Proof* (Sketch). If we know the set of explanations, then we can compute the set of counterexamples (see Remark 1).
Given the set of counterexamples, we can compute one that minimizes some measure of distance to the instance $\mathcal{I}$.
A counterexample represents a cube in feature space; we just need to pick the point closest to $\mathcal{I}$. This can be achieved by fixing the free features. Observe that $\mathcal{I}_{\mathsf{ae}}$ is the complete assignment obtained from the partial assignment $\mathcal{C}_{\mathsf{ae}}$ by setting the missing coordinates to the feature values specified in the given instance $\mathcal{I}$. $\qquad\square$

## 3.3 Exploiting Duality

The duality between absolute explanations $\mathcal{E} \in \mathbb{E}$ and counterexamples $\mathcal{C} \in \mathbb{C}$ for a prediction $\pi$ made by model $\mathbb{M}$ (represented as formula $\mathcal{M}$) can be exploited directly to compute either $\mathbb{E}$, or $\mathbb{C}$, or both. This can be done following the ideas of *prime compilation* of Boolean formulas [44]. Alternatively, the classifier can be compiled into a succinct logical representation, e.g. binary decision diagram (BDD) [52], which allows for efficient enumeration of prime implicants and implicates.

Algorithm 1 shows a Pythonic-style algorithm to compute the complete set $\mathbb{E}$. It computes the set $\mathbb{E}$ of all explanations and a subset of all the counterexamples $\mathbb{C}$. The algorithm utilizes the hitting set duality and represents the *implicit hitting set enumeration* paradigm [9]. It can be seen as a loop, each iteration of which computes a smallest size hitting set $\mathcal{E}$ of the set $\mathbb{C}$ of counterexamples (see

---
**Algorithm 1:** Duality-based computation of all absolute explanations

**Input:**    formula $\mathcal{M}$ and prediction $\pi$
**Output:** set $\mathbb{E}$ of all absolute explanations of prediction $\pi$

```
1  (ℂ, 𝔼, ℰ) ← (∅, ∅, ∅)
2  do:
3      if ℰ ⊨ (𝓜 → π) :
4          𝔼 ← 𝔼 ∪ {ℰ}                        # ℰ is an explanation; save it
5      else:
6          (𝒞, ρ) ← ExtractInstance()  # get an instance 𝒞 with a prediction ρ, ρ ≠ π
7          for l ∈ 𝒞 :
8              if (𝒞 \ {l}) ⊨ (𝓜 → ρ) :
9                  𝒞 ← 𝒞 \ {l}
10         ℂ ← ℂ ∪ {𝒞}                         # update ℂ with a new counterexample 𝒞
11     ℰ ← MinimumHS(ℂ)                         # get a new hitting set of ℂ
12 while ℰ ≠ ∅
13 return 𝔼
```
---

line 11) and checks whether or not $\mathcal{E}$ is an explanation for prediction $\pi$ (line 3). (This check can be done by calling an oracle testing unsatisfiability of formula $\mathcal{E} \wedge \mathcal{M} \wedge \neg\pi$.) If it is, $\mathcal{E}$ is added to $\mathbb{E}$. Otherwise, i.e. if $\mathcal{E} \wedge \mathcal{M} \wedge \neg\pi$ is satisfiable, a satisfying assignment exists defining an instance $\mathcal{C}$ that is classified by $\mathcal{M}$ as some $\rho \in \mathbb{K}$ s.t. $\rho \neq \pi$. Such satisfying assignment is typically easy to obtain from the oracle (line 6). Instance $\mathcal{C}$ is then reduced to a counterexample by removing all *redundant*, i.e. unnecessary, literals (see the loop line 7–line 9 and concretely the oracle call in line 8). The new counterexample is added to the set $\mathbb{C}$ (see line 10) and a new hitting set $\mathcal{E}$ is obtained (line 11). The algorithm proceeds until there is no more hitting set $\mathcal{E}$ of $\mathbb{C}$. Observe that *initially* $\mathcal{E}$ is empty and, thus, the first iteration of Algorithm 1 *always* results in a new counterexample $\mathcal{C}$ being computed and added to $\mathbb{C}$. Note that although Algorithm 1 targets enumerating all explanations $\mathbb{E}$, it can also be applied for computing the set $\mathbb{C}$ of all counterexamples, with minimal modifications of the formulas in line 3 and line 8.

**Remark 3.** *Although the goal of Algorithm 1 is to illustrate a way to exploit the duality, its practical efficiency may not be ideal in some specific settings. The algorithm relies on the oracle calls in line 3 and line 8, which are NP-hard. Also, extracting a smallest size hitting set is NP-hard as well and can be done with the use of modern optimization procedures, e.g. mixed integer programming (MILP) [20] or maximum satisfiability (MaxSAT) [31]. Furthermore, in the worst case, the algorithm could end up enumerating both all explanations and all counterexamples and there might be an exponential number of them. However, in other settings, this worst-case scenario is not often observed in practice.*

## 4  Experimental Evidence

The section practically illustrates the described duality between the concepts of absolute explanation and counterexample for a given model prediction. To do this, the following experiment was performed on a Macbook Pro with an Intel Core i5 2.3GHz CPU and 16GB of memory. The experiment targets the well-known and widely used MNIST digits database[4] as it enables a *visual demonstration* of the discovered duality relationship. As a classifier model, we consider neural networks (NNs) with *rectified linear unit* (ReLU) non-linear activation operators and the known encoding of ReLU-based NNs into MILP [14]. The developed Python-based prototype[5] follows the prime compilation approach of Algorithm 1 and uses CPLEX 12.8.0 [20] as an MILP oracle, which is invoked at each iteration of the algorithm. The implementation of minimum hitting set enumeration of Algorithm 1 is based on an award-winning maximum satisfiability solver RC2[6] [22] written on top of the PySAT toolkit [21].

For the sake of simplicity, the networks used are trained to distinguish two digits, e.g. 5 and 6 (because of their visual resemblance). Also, due to a significant number of explanations and counterexamples,

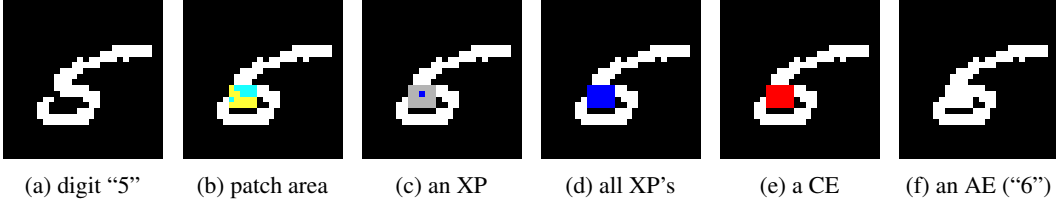

| (a) digit "5" | (b) patch area | (c) an XP | (d) all XP's | (e) a CE | (f) an AE ("6") |

Figure 1: An example of digit five.

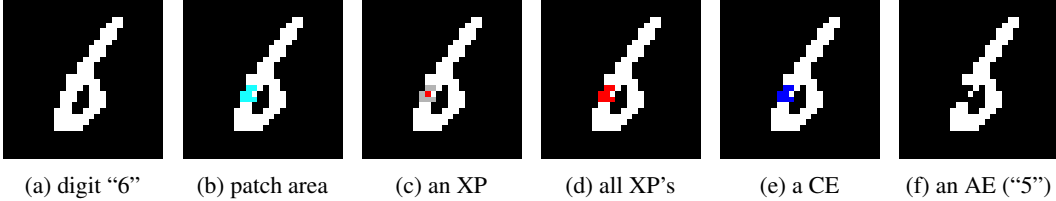

| (a) digit "6" | (b) patch area | (c) an XP | (d) all XP's | (e) a CE | (f) an AE ("5") |

Figure 2: An example of digit six.

the following is assumed: (1) only pixels from a predefined patch area can participate in an explanation/counterexample with the other pixels being fixed; (2) the images were binarized, i.e. every pixel can be either black or white. However, note that the duality holds for the most general case with assumptions (1) and (2) being disabled.

Figure 1a and Figure 2a show two concrete examples of digits 5 and 6. The patch areas for these images are highlighted in Figure 1b and Figure 2b. The patches contains 20 and 7 pixels, respectively. These patch areas are selected intentionally as their pixels are supposed to be crucial for the prediction being "5" or "6". Enumerating all explanations and counterexamples for these images with the given patches result in 20 (7, resp.) unit-size explanations for digit five (six, resp.). An example of one concrete explanation for these images is shown in Figure 1c and Figure 2c, respectively. Recall that the images are binarized; here, the corresponding pixel is blue (red, resp.) if the explanation sets it black (white, resp.) while the other (gray) pixels of the patch may have *any* color and the prediction will still remain as long as the pixels outside of the patch area are fixed. The unions of all explanations are shown in Figure 1d and Figure 2d, respectively. Also, for both images there is a unique counterexample. These are depicted in Figure 1e and Figure 2e. Observe that "polarities" of the pixels in explanations and counterexamples are opposite to each other. This clearly exhibits the described duality between the concepts of absolute explanations and counterexamples.

Note that the only counterexample for prediction "5" sets all pixels white. Such an image is shown in Figure 1f and represents an adversarial example for Figure 1a, i.e. it is classified as "6". A similar observation can be made with respect to digit six. However, in this case the only counterexample sets all patch pixels black. Thus, an adversarial example for digit six is shown in Figure 2f and it is classified as "5".

## 5    Conclusions

Adversarial examples and explanations of ML models are arguably two of the most significant areas of research in ML. This paper shows a tight relationship between the two. Concretely, the paper proposes the dual concept of counterexample, the notion of breaking an explanation or a counterexample, and shows that each explanation must break every counterexample and vice-versa. This property is tightly related with the concept of hitting set duality between diagnoses and conflicts in model-based diagnosis [45], but also with computation of prime implicants and implicates of Boolean functions [51]. The paper also overviews algorithms for computing explanations from counterexamples and vice-versa. Furthermore, the paper shows how adversarial examples can be computed given a reference instance in feature space and counterexample that minimizes the distance to the instance. The experimental evidence illustrates the applicability of the duality relationship between explanations and counterexamples (and adversarial examples). Future work will investigate extensions to the work to target problems of larger scale.

## Footnotes

[1]Since formula $\mathcal{M}$ computes a function, the prediction for *any* instance is unique. It is straightfoward to devise ML models where $\mathcal{M}$ does not compute a function. Such cases are not considered in this paper.

[2]Explanations find a wide range of uses in AI and CS in general. In general, understanding the causes of inconsistency in over-constrained systems of constraints corresponds to explaining the reasons of inconsistency. In a similar fashion, diagnosis of failing systems can also be seen as explaining the reasons for system failure [45].

[3]Given a set $\mathcal{R}$, subset-minimality of a set $\varphi \subseteq \mathcal{R}$ wrt. a predicate $\mathcal{P}$ over set $\mathcal{R}$ means that (a) $\mathcal{P}(\varphi)$ and (b) $\forall (\varphi' \subsetneq \varphi) \, \neg\mathcal{P}(\varphi')$.

[4] http://yann.lecun.com/exdb/mnist/

[5] https://github.com/alexeyignatiev/xpce-duality/

[6] https://maxsat-evaluations.github.io/

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
