[Reviews · NeurIPS 2019]

Reviewer 1



The work is amazingly well-written and easy to follow. The work does a pretty good job of introducing the required logic background for understanding the rest of the work (I was not introduced to the notions and terminology before and I found it very easy to follow). As a researcher in both fields, I find this new direction very much and appreciate the work's significance. However, from a practitioner's point of view, the main question is the applicability of the introduced algorithm. The algorithm requires computations up to the order of the feature space size (which is intractably large even for the simplest problems). I am leaning towards an acceptance score due to its novelty but the actual contribution and its usefulness in practice should become more clear.

Reviewer 2



This paper address an intuition that has been present in the literature for some time, but has not been formalized or published that i am aware of. Many papers hint at the duality between adversarial examples and explanations, but by formalizing these notions and proving the duality between them, the authors make an important contribution to the literature, the paper is quite dense, and spends most of its time on theoretical proofs and justifications for the relationship between adversarial examples. It may find a wider readership if it allocates some more space to introducing the concepts it employs, even such simple things as "subset-minimal" may not be widely known in the adversarial community. The final section focusing on the experimental result making use of a subset of a binarized version of MNIST is compelling, and makes the theoretical work significantly easier to grasp. The running example of the restaurant running problem, also helps illustrate the theorems presented, but the paragraph describing the specifics of the problem is probably not needed. Overall i think this paper is an important contribution to the research literature, but could be made more approachable and subsequently reach a broader audience with some light rewriting with a focus on making more accessible.

Reviewer 3



Overall Comments First I should state upfront that my expertise is not in first order logic, so it is somewhat difficult to assess this paper. I am familiar with the literature on adversarial examples and explanations though. My main high level point in this work is that I am missing a so what, i.e., what is the significance of demonstrating the duality between explanations and adversarial examples. In addition, there has been intense empirical work showing that explanations can be used to craft adversarial examples and adversarial examples might be good instances for producing explanations. Originality The main point of this paper has been demonstrated empirically in prior work. The authors present a set of theorems based on formal logic demonstrating a duality between adversarial examples and explanations. In terms of the theorems, I am not familiar with the formal logic literature , though in looking at citations [21-23], it does seem that these theorems are new. Clarity The work is reasonably well written and free of typos. Several of the key formal logic terms are also defined and clarified. Theorem 1 was clearly stated and the proof clarified in the text. A proof sketch was also provided for theorem 2. Significance In looking at the prior work, citations [21-23], it seems the theorems here are new. In general, there have been previous connections made between explanations and adversarial examples: https://www.aclweb.org/anthology/P18-1176, and https://homes.cs.washington.edu/~marcotcr/acl18.pdf, however these were not formal. In fairness, it is hard for me to assess the significance of this work since it seems like the key insight is bringing the FOL point of view to clarify the relationship between adversarial examples and explanations. Some Issues - The MNIST Example. I sense this is probably the wrong example to use to show the power of your analysis. It seems like MNIST is high-dimensional for the logic based models or decision set type framework. It would've been more powerful for me, if the paper had shown their results on lower dimensional dataset with enough categorical variables to show the power of this work. The current MNIST example feels toyish. - Can the authors further clarify the implications of the duality that they motivate in this work? UPDATE I have read the author rebuttal and feel the authors provided justification for their approach and why this work is important. I support that this work be accepted.

[Author Response · NeurIPS 2019]

We thank the reviewers for the insightful and helpful comments. Minor remarks will be reflected in the text. Individual responses to the questions are included below.

## Reviewer 1

The reviewer raises the very relevant issue of applicability, both theoretical and practical. The paper's framework is general and can be applied to a wide class of problems and ML models. Hence, we agree with the reviewer in that the theoretical applicability is significant. Moreover, we will improve the presentation of experiments, to highlight the practical applicability of the work.

Also, the practical applicability of our work depends on a reasoner for some relevant fragment of first order logic. The same applies to a number of recent works that exploit formal methods. The improvements made to reasoners, e.g. SMT solvers, in recent years offer guarantees that practical applicability will continue to improve.

Furthermore, there is already practical evidence to the relevance and the insights provided by the use of formal methods in ML; some are referenced in our paper.

The duality relationship that our paper reveals will enable researchers to look at adversarial examples and explanations in a new light. This will open new avenues of research, that will foster more efficient exact methods but also better heuristics.

The reviewer is quite right that explanations find important uses in many other settings. We will expand on this.

## Reviewer 2

We thank the reviewer for the suggestions regarding presentation.

We will improve readability of the paper, e.g. by clarifying the concepts that are not widely known in the community, including some of the logic-related notation used.

The proof of Theorem 1 will be cleaned up in the revised paper, as suggested by the reviewer. All the minor issues spotted will be addressed. Thank you!

## Reviewer 3

The reviewer raises a *'so what'* question.

The importance of having a deep understanding of the connection between adversarial examples and explanations is epitomized by a number of recent works, some of which are cited in our paper.

Also significant is a recent keynote talk by I. Goodfellow at the AAAI 2019 conference on *"Adversarial Machine Learning"* (please see the official video recording starting from 53m55s), which makes a very strong case for relating adversarial examples and explanations. Namely, Goodfellow conjectures that there should be a connection between adversarial attacks and explanations and points out that this connection has not been revealed so far. We believe that our work is a significant step forward in this direction.

We chose an image dataset as we can visually demonstrate the discovered relationship between adversarial examples and explanations to the reader. We want to convey that each explanation hits all adversarial examples and vice versa. This "hitting" process can be highlighted using coloring in the image (please see Figures 1 and 2 in the paper). While our method can be applied to categorical datasets, we feel that such visualization might be less visually appealing to a reader.

[Meta-Review · NeurIPS 2019]

Reviewers all agree that the paper presents interesting theoretical relations between adversarial examples and explanations and is well-written. There are some suggestions on how to provide more intuitive explanations (no pun intended..) for proofs among other things. Please carefully incorporate feedback to your final version.